# Podocyte-Related Mechanisms Underlying Survival Benefit of Long-Term Angiotensin Receptor Blocker

**DOI:** 10.3390/ijms23116018

**Published:** 2022-05-27

**Authors:** Xuejing Zhu, Dan Gao, Vittorio Albertazzi, Jianyong Zhong, Li-Jun Ma, Liping Du, Yu Shyr, Valentina Kon, Hai-Chun Yang, Agnes B. Fogo

**Affiliations:** 1Department of Nephrology, Second Xiangya Hospital, Central South University, Changsha 410011, China; zhuxuejing5225209@163.com; 2Department of Pathology, Microbiology and Immunology, Vanderbilt University Medical Center, Nashville, TN 37232, USA; gaodan1106@sina.com (D.G.); albertazzi_vittorio@hotmail.com (V.A.); jianyong.zhong@vanderbilt.edu (J.Z.); lijunma9999@gmail.com (L.-J.M.); haichun.yang@vumc.org (H.-C.Y.); 3Department of Nephrology, The First Affiliated Hospital, Zhengzhou University, Zhengzhou 450052, China; 4Unit of Nephrology and Dialysis, “Guglielmo da Saliceto” AUSL Piacenza Hospital, Via Taverna 49, 29100 Piacenza, Italy; 5Division of Pediatric Nephrology, Vanderbilt University Medical Center, Nashville, TN 37232, USA; valentina.kon@vumc.org; 6Center for Quantitative Sciences, Vanderbilt University Medical Center, Nashville, TN 37232, USA; liping.du@vanderbilt.edu (L.D.); yu.shyr@vanderbilt.edu (Y.S.); 7Division of Nephrology, Vanderbilt University Medical Center, Nashville, TN 37232, USA

**Keywords:** ARB, glomerulosclerosis, survival, podocyte, proteinuria

## Abstract

We previously found that short-term treatment (week 8 to 12 after injury) with high-dose angiotensin receptor blocker (ARB) induced the regression of existing glomerulosclerosis in 5/6 nephrectomy rats. We therefore assessed the effects of long-term intervention with ARB vs. nonspecific antihypertensives in this study. Adult rats underwent 5/6 nephrectomy and renal biopsy 8 weeks later. The rats were then divided into three groups with equivalent renal function and glomerular sclerosis and treated with high-dose losartan (ARB), nonspecific antihypertensive triple-therapy (TRX), or left untreated (Control) until week 30. We found that blood pressure, serum creatinine levels, and glomerulosclerosis were lower at sacrifice in ARB and TRX vs. Control. Only ARB reduced proteinuria and maintained the density of WT-1-positive podocytes. Glomerular tufts showed more double-positive cells for CD44, a marker of activated parietal epithelial cells, and synaptopodin after ARB vs. TRX or Control. ARB treatment reduced aldosterone levels. ARB-treated rats had significantly improved survival when compared with TRX or Control. We conclude that both long-term ARB and triple-therapy ameliorate progression, but do not sustain the regression of glomerulosclerosis. ARB resulted in the superior preservation of podocyte integrity and decreased proteinuria and aldosterone, linked to increased survival in the uremic environment.

## 1. Introduction

Glomerulosclerosis induced by nephron loss is a common feature of chronic kidney disease (CKD), characterized by increased extracellular matrix (ECM) causing glomerular tuft obliteration, regardless of the primary disease. The rate of progressive loss of glomerular filtration rate (GFR) differs according to the specific kidney disease and also among individual patients with the same disease. The regression of chronic injury is possible. Thus, ten years after the pancreas transplantation that cured type I diabetes, diabetic kidney sclerosis was reduced [1]. In the African-American Study of Kidney Disease (AASK) trial and NephroTest NEPHROTEST study, some patients even exhibited increased GFR over time [2,3]. Angiotensin converting enzyme (ACE) inhibitor or angiotensin II receptor blocker (ARB) ameliorate proteinuria and induce regression beyond effects on systemic blood pressure in CKD [4,5,6]. The Ramipril Efficacy in Nephropathy (REIN) follow-up study showed that many patients on continued ramipril therapy even had improved GFR [7]. We and others used high-dose ARB from week 8 to 12 after injury to induce the regression of existing glomerulosclerosis in the 5/6 nephrectomy rat model [8,9,10]. We then assessed whether the regression of glomerular sclerotic lesions can be sustained long-term.

The regression of glomerulosclerosis encompasses a decreased matrix accumulation and an increased glomerular capillary area. In our previous study, the regression induced by ARB was linked to reduced plasminogen activator inhibitor-1 (PAI-1) and increased matrix degradation [8]. The regeneration of the glomerular capillary network requires endothelial cell proliferation and angiogenesis. Podocytes not only support capillary integrity but also stimulate angiogenesis by secreting VEGF-A. Podocytes have limited regeneration capacity, and podocyte loss is key for progressive glomerulosclerosis. Potential stem cells contributing to podocyte restoration include interstitial progenitor cells, renin positive cells, and parietal epithelial cells (PECs). The administration of ACEI, ARB, retinoids, or an improved diabetic milieu has been shown to increase the glomerular density of podocyte markers following an initial phase of depletion [11,12,13,14]. However, whether the regeneration of podocytes can result in less glomerulosclerosis and whether ARBs can accelerate this process long-term is unknown.

CKD dramatically shortens life span, especially due to the increased incidence of cardiovascular disease. Interestingly, a remarkable prolongation of life span was observed in mice deficient in AT1a angiotensin receptor [15]. A polymorphism in the AT1 gene promoter that blunts its function has been associated with extreme human longevity [16]. We therefore assessed long-term ARB effects on glomerulosclerosis and mortality in a rat model of CKD.

We show that in contrast to our previous short-term studies, long-term monotherapy with ARB does not sustain the regression of glomerulosclerosis, but rather slows progression, at a similar rate as with TRX treatment. Nonetheless, compared to TRX, ARB is superior in maintaining podocytes, reducing proteinuria and plasma aldosterone, which we suggest underlies the ARB effects, to significantly increase survival in CKD.

## 2. Results

### 2.1. Long-Term ARB or TRX Effects on Systemic and Renal Parameters Subsection

Blood pressure increased significantly from week 0 to 8 after 5/6 Nx in all groups and continued to increase from week 8 to sacrifice in Control, but decreased in ARB and TRX. The rate of change in systemic blood pressure over time was lower in ARB vs. Control and was even lower in TRX (Table 1 and Figure 1A).

Twenty-four-hour urinary protein excretion was similarly increased in all groups at week 8 before treatment, by study design. Although, proteinuria continued to increase in Control and TRX and was stabilized by ARB. Thus, from biopsy (Bx) to sacrifice, TRX had a significantly greater increase in proteinuria than ARB (*p* < 0.05, Table 1 and Figure 1B).

GFR, assessed by 24 h creatinine clearance, decreased from week 0 to 8 after 5/6 Nx in all groups. From week 8 to the endpoint, GFR was maintained in ARB and Control, and decreased further in TRX (Table 1 and Figure 1C).

We next assessed urinary KIM-1, a functional marker for tubular injury. TRX significantly reduced the rate of decline in urinary KIM-1 (*p* < 0.05 vs. Control), whereas ARB only numerically decreased this injury marker (Table 1 and Figure 1D).

### 2.2. Both ARB and TRX Reduced Glomerulosclerosis but Not Interstitial Fibrosis

By study design, all groups had similar average glomerulosclerosis severity at biopsy. From biopsy to sacrifice, the glomerulosclerosis severity progressed in Control (Table 1 and Figure 2A). In contrast, there was less progression of glomerulosclerosis in ARB and TRX (*p* < 0.05 vs. Control, Figure 2A,B). Only 1 of 20 rats treated with ARB showed regression, i.e., less sclerosis at autopsy vs. biopsy, similar to Control (1/21) and TRX (1/20). Interstitial fibrosis, assessed by Sirius red staining, was similar at biopsy in all groups (Table 1). The progression of interstitial fibrosis at sacrifice was not different among groups (Figure 2C,D).

Glomerular macrophage infiltration was significantly reduced by ARB but only numerically reduced by TRX vs. Control. There was no difference in interstitial inflammation among the three groups (Appendix A).

### 2.3. ARB, but Not TRX, Maintained WT-1^+^ Cell Density

WT-1^+^ cell density was assessed as a marker of mature podocytes in glomeruli. WT-1^+^ cell density was similar at biopsy in the three groups and decreased in all groups at the time of sacrifice (Table 1). However, the reduction of WT-1^+^ cell density was significantly less in ARB-treated rats compared to Control and TRX groups (Figure 3A).

CD44 is a marker for activated PECs, which can line Bowman’s capsule or can be found on the glomerular tuft. Some CD44^+^ cells on the tuft also expressed synaptopodin, indicating a possible PEC-podocyte transition (Figure 3B). Such double-positive cells were only rarely seen at biopsy. However, ARB treatment resulted in more cells with CD44/synaptopodin double staining on the tuft than in TRX or Control groups (Table 1 and Figure 3B).

### 2.4. Effects of ARB and TRX on the Heart

ARB treatment caused a numerical reduction in myocyte hypertrophy by 28.4% and heart fibrosis by 52.1% vs. Control, whereas these parameters in TRX were similar to Control (Figure 4A). Plasma aldosterone was significantly increased from biopsy to autopsy in both Control and TRX groups, with significantly less increase in the ARB group (Figure 4B). Klotho, a factor produced by the renal tubular epithelial cells with numerous effects that include protecting the heart against stress-induced cardiac hypertrophy and remodeling, was not different in ARB, TRX or Control groups at sacrifice (Control 17.2 ± 4.18, ARB 16.5 ± 2.55, TRX 12.0 ± 2.58 ng/mL) [17]. Macrophage infiltration, detected by the CD68 staining of the heart, was reduced by ARB but not by TRX treatment, compared to Control (Figure 4C).

### 2.5. Long-Term ARB, but Not TRX Increased Survival

Survival was significantly increased in ARB-treated rats compared to either the Control or TRX group (Figure 5A). Log-rank tests were used for this comparison, which indicates that the *p* value is 0.006 for Control vs. ARB and *p* value is 0.03 for TRX vs. ARB. Survival was not different in TRX vs. Control rats. By week 30, 18/21, and 18/20 rats died in Control and TRX groups, but only 11/20 ARB rats died. The median survival time after 5/6 Nx in Control was 19.7 weeks, and 25.1 weeks in TRX. By contrast, the median survival time increased to 27.8 weeks in ARB.

We next assessed the various factors which could influence survival. In Control, severe glomerulosclerosis at the onset of therapy increased risk (HR = 4.86, 95% CI 1.48–15.90, when starting sclerosis index (0–4 scale) increased from 0.78 to 1.83). By contrast, other factors, including the level of blood pressure and proteinuria at the onset of therapy, did not significantly affect survival in these untreated control rats (Figure 5B). ARB, but not TRX, reduced the risk of death (hazard ratio to control, HR = 0.26, 95% CI 0.08–0.83) when comparing rats with the same starting glomerulosclerosis index (SI), systolic BP (SBP), and proteinuria (Figure 5B). Furthermore, the benefit associated with ARB treatment on survival relative to the control was independent of the glomerulosclerosis at the onset of treatment, with similar HRs across levels of sclerosis (Figure 5C).

## 3. Discussion

In this long-term CKD study, we found that both ARB and TRX prevented the progression of glomerulosclerosis. However, these beneficial effects may be through different mechanisms. Both TRX and ARB have similar protection on glomerulosclerosis after 5/6Nx, but podocyte density was higher in ARB than in TRX, linked to less progression of proteinuria. The progression of glomerulosclerosis has been linked to a decreased repair response of podocyte progenitor/stem cells of extrarenal or renal origin after injury. Human renal cell progenitors isolated from the urinary pole of Bowman’s capsule can differentiate into podocytes or tubular cells in vitro and can induce tubular regeneration when infused into mice with acute kidney injury [18]. In both experimental models and patient studies, ACEI therapy sustained glomerular repair by modulating progenitor cell proliferation, limiting crescent generation, and promoting podocyte repair [19,20]. ARB has also been shown to improve podocyte regeneration by augmenting the number of parietal epithelial cell progenitors. In this study, ARB treatment resulted in improved WT-1 positive cell density in glomeruli compared to Control and TRX treatment. Our results also showed increased glomerular tuft cells positive for markers for both active PECs and podocytes after long-term ARB compared to Control and TRX. These results support that ARB has beneficial effects on glomerular epithelial cells via inducing a transition from PECs to podocytes. TRX significantly decreased urine KIM-1 levels, indicating that TRX treatment may have a better protective effect on tubular function than ARB. Hydrochlorothiazide, one of the drugs in this TRX, has an anti-proliferative effect on tubular cells and can reduce albumin reabsorption by tubular epithelial cells [21]. Thus, TRX may have beneficial effects on glomerular injury by decreasing tubular injury through tubuloglomerular crosstalk.

A significant finding in our study was the prolonged survival time induced by ARB. This survival benefit was present regardless of the severity of glomerular injury at the beginning of treatment. These results parallel clinical findings that show that ARB reduced the risk of all-cause mortality by 24.6% in patients with hypertension and CKD [22]. In addition, although TRX treatment resulted in lower blood pressure than ARB in the current study, TRX did not improve the survival rate, indicating that this beneficial effect of ARB involves mechanisms other than systemic BP reduction. ARB induced less progression of proteinuria and GFR than TRX. These results parallel clinical studies showing that treatments that effectively decrease proteinuria have beneficial effects on GFR, as well as on the excretion of uremic toxins. Indeed, by multivariable analysis, proteinuria reduction was the only variable associated with the lessened rate of GFR decline and less risk for ESKD in patients [23]. A meta-analysis showed that the risk of ESKD decreased by 23.7% for each 30% reduction in albuminuria [24]. In addition to greater progression to ESKD, patients with significant residual proteinuria after treatment (≥1 g/24 h) also had more cardiovascular events [25]. The pathophysiological mechanisms linking albuminuria to cardiovascular risk are not well understood. Albuminuria largely reflects glomerular damage and is considered a marker of systemic vascular damage [26,27]. Albumin in the urine can directly damage the kidney, increasing tubular injury and proinflammatory and profibrotic factors. Currently, standard markers of kidney function and structural injury, including eGFR, glomerulosclerosis, and interstitial fibrosis are insensitive and do not adequately capture the range of key injuries. Thus, despite little measurable change in serum creatinine or GFR, key parameters that affect permeability and other renal and systemic effects may be affected by ARB. These include effects on the heart or the uremic environment, tubular function, or quality and amount of proteinuria. Similarly, although glomerulosclerosis and interstitial fibrosis were not different in ARB and TRX groups, a more detailed analysis showed the superior maintenance of WT-1 positive cells, a marker of differentiated podocytes, with ARB treatment. Improved podocyte density may improve subtle aspects of renal function not adequately captured by eGFR/creatinine.

The increased survival of ESKD patients treated with ARB may also be related to its extrarenal effects, such as cardiovascular protection [28]. Plasma aldosterone was decreased in the ARB group compared to TRX, associated with significantly reduced inflammation, and numerically less myocyte hypertrophy and heart interstitial fibrosis in ARB compared to Control and TRX, albeit with only limited samples evaluated. Elevated plasma aldosterone levels are associated with increased mortality in patients with severe heart failure [29]. Suppressing aldosterone production in the adrenal requires the blockade of both G protein-dependent and β-arrestin-1-dependent pathways [30]. Losartan almost completely blocks the AT1R-G-aldosterone pathway, whereas EXP1374, an active metabolite of losartan, effectively suppresses β-arrestin-1 [31]. Recent clinical trials indicated that finerenone, an aldosterone antagonist, improved cardiovascular outcomes in patients with type 2 diabetes and CKD with albuminuria [32].

In the current study, neither the ARB nor nonspecific antihypertensive triple therapy treatment affected the long-term regression of existing glomerulosclerosis. Previous studies in human CKD have shown that multipronged therapy, including maximum proteinuria reduction by titrated dual renin-angiotensin-system (RAS) inhibition, amelioration of dyslipidemia with statins, smoking cessation, lowering salt intake, and healthy lifestyle implementation, can regress CKD [23,33]. Similarly, we and others have found that a combination of aldosterone synthase inhibitor and ARB, or a combination of proliferator-activated receptor γ (PPARγ) agonist and ARB, induced more regression of glomerulosclerosis than monotherapy in the 5/6 nephrectomy model [34]. These additional protective effects were related to less inflammation and more podocyte protection. In a recent meta-analysis study, sodium-glucose cotransporter 2 (SGLT2) inhibitor in combination with an RAS blocker induced the additional reduction of albuminuria, creatinine, cardiovascular mortality, and heart failure-related hospitalization rates vs. RAS blocker alone [35]. Taken together, these findings strengthen the concept that once set in motion, the mechanisms involved in CKD cannot be controlled by simply inhibiting the RAS and may require a multi-pronged strategy through novel anti-inflammatory and antifibrotic molecules.

In conclusion, both ARB and triple therapy reduced the progression of existing glomerulosclerosis long-term, but neither one sustained the regression of glomerulosclerosis. Nonetheless, ARB, but not triple therapy, enhanced podocyte protection, reduced aldosterone, and caused a striking increase in survival. Our data suggest that drugs targeting podocytes may enhance survival in the uremic environment through beneficial effects on permeability functions.

## 4. Materials and Methods

### 4.1. Experimental Design and Animals

Adult male Sprague Dawley rats (250 to 300 g; Charles River, Wilmington, MA, USA) were used and housed under normal conditions with a 12-h light/dark cycle, at 70 °F with 40% humidity and received normal rat chow and water ad libitum. 5/6 Nx was done by right unilateral Nx and ligation of branches of the left renal artery, producing a total of 5/6 renal ablation. Rats underwent open renal biopsy at week 8, as previously described, and were divided into the following three groups with an equal average starting glomerulosclerosis index (SI): Control animals received no further treatment (Control, *n* = 21); ARB group (ARB, *n* = 20) received 200 mg/L losartan in drinking water; triple-therapy group (TRX, *n* = 20) received antihypertensive drugs (reserpine 5 mg/L, hydralazine 80 mg/L, hydrochlorothiazide 25 mg/L) in drinking water [8]. This ARB dose was used in our previous study, showing short-term regression of glomerulosclerosis over 4 weeks [8]. This TRX dose has been previously shown to decrease systemic blood pressure in this model [36]. Treatments continued until week 30 after 5/6 Nx or when rats died or were sacrificed due to advanced disease.

### 4.2. Analysis of Systemic Parameters and Renal Function

Systolic BP (SBP) and proteinuria were assessed at weeks 0, 8, 16, 24, and 30. Tail-cuff plethysmography (IITC; Life Science Inc., Woodland Hills, CA, USA) was used for SBP measurement in trained unanesthetized rats. Animals were placed in metabolic cages for 24 h for urine collection, and urine protein level was measured by Bio-Rad Protein Assay Kit (Bio-Rad Laboratories, Hercules, CA, USA). Serum and urine creatinine was measured by Vitros CREA slides (Johnson & Johnson Clinical Diagnostics Inc., Rochester, NY, USA). GFR was calculated by creatinine clearance.

### 4.3. Morphology Analyses

Kidney tissue from biopsy and autopsy was immersion-fixed in 4% paraformaldehyde/PBS and routinely processed, and 3 μm sections were stained with periodic acid-Schiff. Biopsy samples contained on average 23 glomeruli (range in each group, 19 to 35). Autopsy sections for analysis contained >100 glomeruli, on average.

Glomerulosclerosis was evaluated by a semi-quantitative score. Each glomerulus was graded from 0 to 4+ as follows: 0, no lesion; 1+, sclerosis of <25% of the glomerulus; 2+, 3+, and 4+, sclerosis of 25 to 50%, 50 to 75%, and ≥75% of the glomerulus, respectively. The final score of each rat was the average score from all glomeruli in one section [8].

Tubulointerstitial fibrosis was quantitated by Sirius red staining. Briefly, slides were stained in saturated picric acid with 0.1% Sirius Red F3BA (Sigma-Aldrich Co., LLC, Saint Louis, MO, USA) overnight, followed by 2 min washing in 0.01 N hydrochloric acids. Polarized 40× images from the cortex area were acquired, and the percentage of the positive area was analyzed by using Image J software.

Heart tissue was fixed in 4% paraformaldehyde/PBS solution and embedded in paraffin. Sections (3 μm) were cut and stained with hematoxylin and eosin or Masson’s trichrome. Cardiac fibrosis areas were measured on trichrome staining using an image analysis system (KS 400 Imaging System; Carl Zeiss Vision, Eching, Germany). Myocyte cross-sectional area was used for the evaluation of the degree of ventricular hypertrophy [37].

### 4.4. Immunohistochemistry Staining

Paraformaldehyde–fixed, paraffin-embedded sections were used for immunostaining. For single and double staining, paraffin sections were rehydrated and antigen was retrieved by microwave. Sections were incubated with the following primary antibodies at 4 °C overnight: rabbit anti-WT-1 (1:200, RB-9209-P1; Santa Cruz Biotechnology, Santa Cruz, CA, USA), mouse anti-synaptopodin (1:100, sc-50459; Santa Cruz Biotechnology), rabbit anti-CD44 (1:200, 65,294; Progen, Brisbane, Australia), mouse anti-CD68 (1:500, MCA341R; Bio-Rad Laboratories). The tissue was incubated with secondary antibodies for 30 min at RT. Slides were counterstained with hematoxylin or Hoechst 33,258 (Molecular Probes, Darmstadt, Germany) to stain nuclei. WT-1 positive density was expressed as the number of positive cells in each glomerulus divided by the glomerular tuft area. The percentage of double-positive cells (CD44+ and synaptopodin+) per CD44+ cells was calculated, and CD68 positive density was expressed as percentage of area. Positive and negative controls stained appropriately. All tissue assessments were made while unaware of treatment group.

### 4.5. ELISA Assay

Urinary kidney injury molecule (KIM-1) (R&D Systems, Minneapolis, MN, USA), plasma Klotho (Abbexa Ltd., Cambridge, UK), and aldosterone levels (Abcam, Cambridge, MA, USA) were measured by ELISA kits according to the manufacturer’s instructions.

### 4.6. Statistical Analysis

Rats died at different time points. We therefore analyzed results with absolute value at different time points (8 weeks vs. sacrifice) and also by change adjusted by time (Δ/week, rate of change). Data are expressed as mean ± SD (standard deviation) in Table 1. Statistical difference was assessed by a single-factor ANOVA followed by Bonferroni’s correction as appropriate. Nonparametric Kruskal–Wallis tests were also used. The association of treatments with survival was assessed using the Log-rank test and the median survival times were estimated using the Kaplan–Meier method. The adjusted hazard ratio (HR) of treatment was also estimated by a Cox regression model. The proportional hazards assumption was tested using Schoenfeld residuals. The effect of SI on survival for each treatment group was based on a Cox regression model with robust standard errors. The effect of SI was modeled with a restricted cubic spline function (non-linear function), and it was also interacted with the treatment group in the model. A *p*-value of less than 0.05 was considered to be statistically significant. All analyses were performed using the software R (https://www.R-project.org/ (accessed on 20 March 2017)) and related packages.

## Figures and Tables

**Figure 1 ijms-23-06018-f001:**
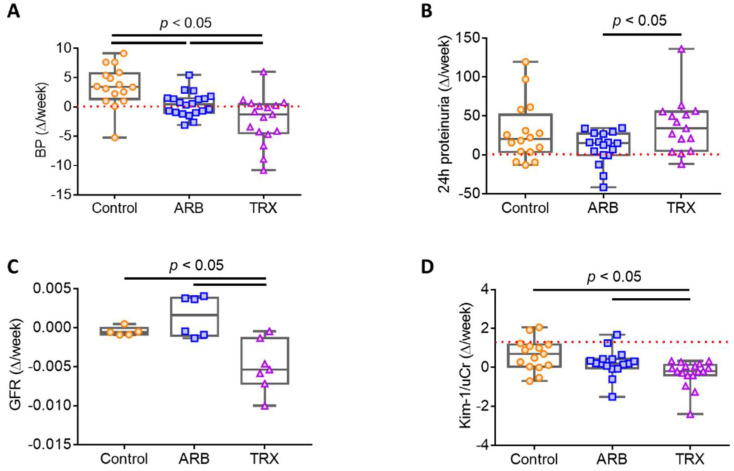
Long-term ARB and TRX treatment had different effects on renal function. (**A**) Changes in blood pressure (BP) from biopsy to autopsy. Each point represents the change in BP from biopsy (Bx) to autopsy (Ax) in an individual rat. The rate of BP increase was less in ARB and TRX compared to Control. (**B**) Changes in proteinuria from biopsy to autopsy. TRX had more proteinuria than ARB. (**C**) GFR decreased more from biopsy to autopsy in TRX compared to Control and ARB. (**D**) Changes in urine Kim-1 from biopsy to autopsy (normalized by urine creatinine, Cr). The change of urine Kim-1 in TRX was less than in ARB vs. Control (*p* < 0.05).

**Figure 2 ijms-23-06018-f002:**
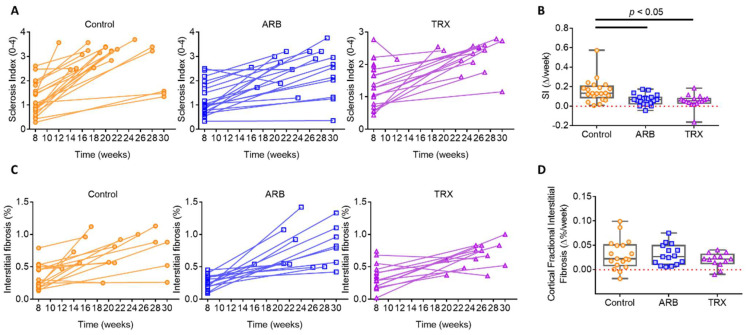
Long-term ARB and TRX reduced glomerulosclerosis but not interstitial fibrosis. (**A**,**B**) Changes in sclerosis index (SI, 0-4 scale) from biopsy to autopsy. SI increased in untreated control 5/6 Nx rats from biopsy to autopsy, with less increase in ARB and TRX. (**C**,**D**). Changes in interstitial fibrosis (Sirius red) from biopsy to autopsy were similar among groups (*p* > 0.05).

**Figure 3 ijms-23-06018-f003:**
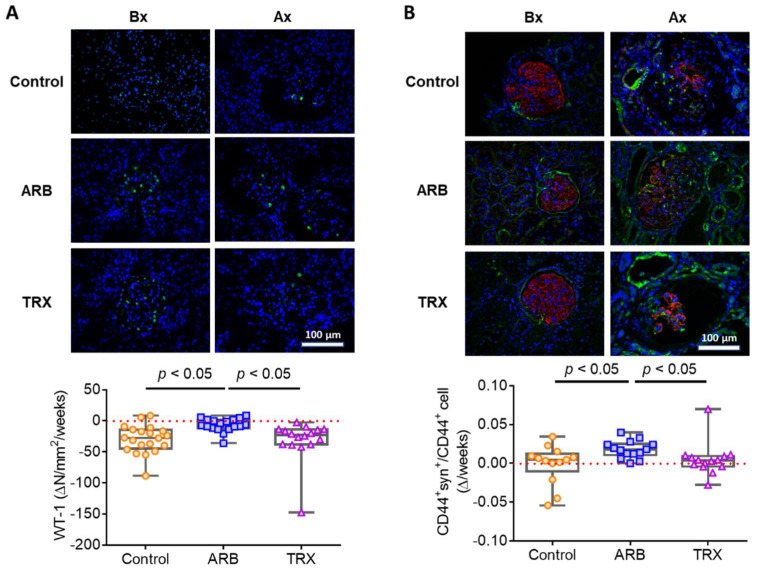
Long-term ARB, but not TRX, maintained glomerular WT-1^+^ cell density, with possible PEC-podocyte transition. (**A**) Immunofluorescence staining for WT-1 (green), representing mature podocytes (arrows, positive cells, blue nucleus, X400). ARB resulted in more preserved glomerular WT-1 positive cell density (*p* < 0.05 vs. Control and TRX). (**B**) Double immunofluorescence staining for CD44 and synaptopodin. CD44 is a marker of activated PECs (green), and synaptopodin stains podocytes (red) (arrows, double-positive cells, blue nucleus, ×400). ARB increased double-positive cells on the glomerular tuft (*p* < 0.05 vs. Control and TRX).

**Figure 4 ijms-23-06018-f004:**
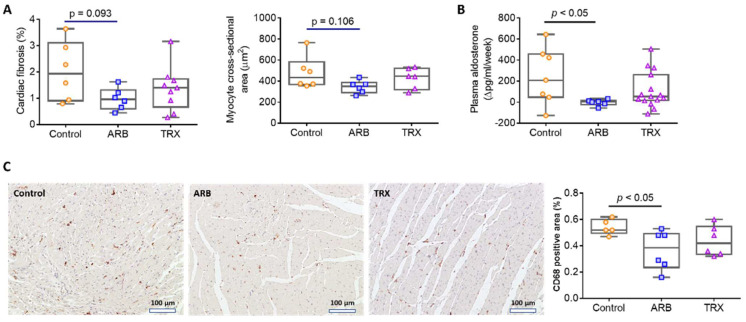
Long-term ARB and TRX treatment and cardiovascular parameters. (**A**) ARB numerically reduced myocyte hypertrophy and heart fibrosis compared to the control group. (**B**) ARB ameliorated the increase in plasma aldosterone level compared to Control (*p* < 0.05). (**C**) Infiltrating macrophages were reduced by ARB but not by TRX treatment, vs. Control (×200).

**Figure 5 ijms-23-06018-f005:**
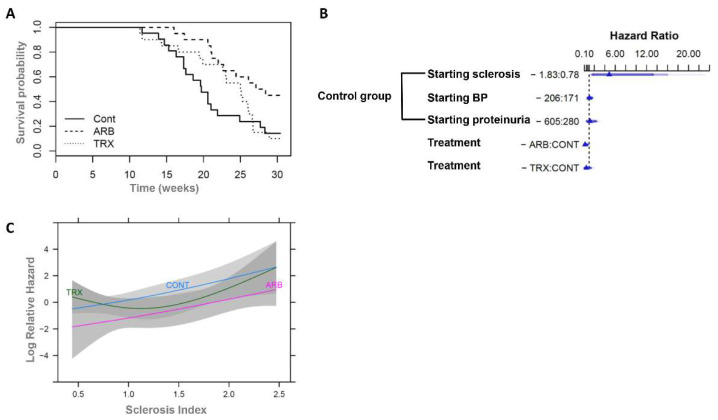
Long-term ARB, but not TRX, increased survival in the rat 5/6 Nx model. (**A**) Kaplan–Meier curves of survival time, and treatment effects. More rats survived with ARB compared to TRX or Control (Log-rank test: Control vs. ARB, *p* < 0.01; TRX vs. ARB, *p* < 0.05; Control vs. TRX, *p* = 0.5). (**B**) Hazard ratios in Cox proportional hazards model. The ARB treatment hazard ratio is 0.26, whereas in untreated Control rats, the hazard ratio related to biopsy glomerular sclerosis severity is 4.86. (**C**) The benefit on survival associated with ARB treatment was independent of starting sclerosis level. The effect of SI on survival was based on a Cox regression model, and effect of SI and its interaction with the treatment groups modeled as described in methods.

**Table 1 ijms-23-06018-t001:** Key functional and structural data at biopsy (8 week) and end time point (up to 30 week).

	Cont		ARB		TRX	
	Biopsy	End Time Point	Biopsy	End Time Point	Biopsy	End Time Point
BP (mmHg)	189.8 ± 17.0 (*n* = 21)	230.9 ± 27.2 (*n* = 16)	182.4 ± 23.9 (*n* = 20)	187.2 ± 22.7 (*n* = 20) *	203.7 ± 34.9 (*n* = 19) ^#^	170.7 ± 35.2 (*n* = 17) *
BW (g)	389.9 ± 37.5 (*n* = 21)	424.1 ± 52.4 (*n* = 16)	390.4 ± 40.4 (*n* = 20)	440.6 ± 41.2 (*n* = 20)	371.0 ± 35.0 (*n* = 19) ^#^	353.9 ± 38.2 (*n* = 17) *^#^
Proteinuria (mg/day)	533.8 ± 200.2 (*n* = 19)	782.2 ± 283.2 (*n* = 18)	430.8 ± 291.6 (*n* = 19)	572.5 ± 252.1 (*n* = 18) *	398.4 ± 180.8 (*n* = 19)	834.6 ± 327.2 (*n* = 16) ^#^
GFR (ml/min/100 g BW)	0.13 ± 0.04 (*n* = 8)	0.15 ± 0.03 (*n* = 5)	0.11 ± 0.04 (*n* = 6)	0.16 ± 0.04 (*n* = 9)	0.18 ± 0.06 (*n* = 18) ^#^	0.12 ± 0.08 (*n* = 7)
Glomerulosclerosis index (0–4)	1.22 ± 0.66 (*n* = 21)	2.88 ± 0.70 (*n* = 21)	1.23 ± 0.66 (*n* = 20)	2.37 ± 0.86 (*n* = 20) *	1.41 ± 0.65 (*n* = 20)	2.28 ± 0.44 (*n* = 16) *
Interstitial fibrosis (%)	0.36 ± 0.20 (*n* = 18)	0.72 ± 0.25 (*n* = 21)	0.26 ± 0.10 (*n* = 18)	0.80 ± 0.32 (*n* = 15)	0.35 ± 0.19 (*n* = 17)	0.69 ± 0.22 (*n* = 14)
Glomerular WT-1 density (N/mm^2^)	496.2 ± 192.7 (*n* = 20)	139.3 ± 82.5 (*n* = 21)	429.9 ± 136.6 (*n* = 18)	311.4 ± 139.4 (*n* = 20) *	510.3 ± 149.7 (*n* = 20)	145.2 ± 85.4 (*n* = 16) ^#^
CD44^+^/Synaptopodin^+^ (N/glomerulus)	0.13 ± 0.19 (*n* = 16)	0.15 ± 0.10 (*n* = 18)	0.12 ± 0.11 (*n* = 18)	0.42 ± 0.20 (*n* = 15) *	0.09 ± 0.15 (*n* = 20)	0.16 ± 0.11 (*n* = 14) ^#^
Urinary Kim-1 (pg/mg)	11.58 ± 4.99 (*n* = 16)	16.58 ± 6.92 (*n* = 16)	10.41 ± 6.01 (*n* = 16)	13.67 ± 6.38 (*n* = 16)	12.11 ± 6.76 (*n* = 19)	9.83 ± 2.54 (*n* = 18) *
Serum aldosterone (pg/mL)	427.7 ± 381.5 (*n* = 7)	2691.0 ± 2159.0 (*n* = 7)	442.6 ± 315.4 (*n* = 6)	432.5 ± 270.3 (*n* = 6) *	448.5 ± 355.7 (*n* = 19)	2295.0 ± 2801.0 (*n* = 15) ^#^

* *p* < 0.05 vs. Control; ^#^ *p* < 0.05 vs. ARB.

## Data Availability

Not applicable.

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
