# Peer review of "Podocyte-Related Mechanisms Underlying Survival Benefit of Long-Term Angiotensin Receptor Blocker"

_ijms, 2022, doi:10.3390/ijms23116018_

Round 1

Reviewer 1 Report

The manuscript is well-written, with clear presentation of the data and neat colclusions. I only have a few minor comments:

Line nos. 154 - 155: 'Survival was significantly increased in ARB-treated rats compared to either Control or TRX group (Figure 5A)' - please provide statistical tests for this claim.

Figure 5C is not clear to me. What is this plot showing? What are the fits? Is it regression? If so, what kind, since the fit for TRX seems to be some sort of a polynomial function. 

Line 150 - 151: 'Inflammation, detected by CD68 staining of heart, was reduced by ARB, but not by TRX treatment, compared to Control (Figure 4C)'. Inflammation is a very general term that involves many different cells types and soluble factors. Calling macrophage infiltration inflammation is an oversimplification. Therefore, I would suggest the authors to remove the term inflammation and call the phenomenon as it is, i.e. macrophage infiltration.

Author Response

We thank the reviewers and editors for their positive and helpful comments.

1)      Line nos. 154 - 155: 'Survival was significantly increased in ARB-treated rats compared to either Control or TRX group (Figure 5A)' - please provide statistical tests for this claim.

We have indicated these details in Figure 5 legend. Log-rank tests were used for this comparison which indicates that the p value is 0.006 for Control vs. ARB and p value is 0.03 for TRX vs. ARB, we have added these specific values to the paper.

2)      Figure 5C is not clear to me. What is this plot showing? What are the fits? Is it regression? If so, what kind, since the fit for TRX seems to be some sort of a polynomial function.

Figure 5C shows the effect of SI on survival for each treatment group (Control, ARB and TRX). It was based on a Cox regression model with robust standard errors. The SI was modeled with a restricted cubic spline function (non-linear function) and it was also interacted with the treatment group in the model. The plot indicates that ARB has better survival benefit than Control and TRX no matter the starting glomerulosclerosis level. This has been added to the Methods and Figure 5C legends.

3)      Line 150 - 151: 'Inflammation, detected by CD68 staining of heart, was reduced by ARB, but not by TRX treatment, compared to Control (Figure 4C)'. Inflammation is a very general term that involves many different cells types and soluble factors. Calling macrophage infiltration inflammation is an oversimplification. Therefore, I would suggest the authors to remove the term inflammation and call the phenomenon as it is, i.e. macrophage infiltration.

We thank the reviewer for this highly relevant comment. We have replaced “inflammation” by “macrophage infiltration”.

Reviewer 2 Report

Dear Editor and authors,

I want to congratulate the authors for this original research dealing with a very important subject, the beneficial efect of ARB upon glomerular histology and survival in CKD. The paper is very well written and easy to understand, the results are clearly presented. I recommend the methodology chapter to be introduced after intoduction, where it is normally placed in a scientific paper.

Author Response

I want to congratulate the authors for this original research dealing with a very important subject, the beneficial efect of ARB upon glomerular histology and survival in CKD. The paper is very well written and easy to understand, the results are clearly presented. I recommend the methodology chapter to be introduced after intoduction, where it is normally placed in a scientific paper.

We thank the reviewer and editor for their positive and helpful comments.

Per journal guidelines, we placed the Methods section after the Discussion.
